# Amyloid Aβ_25-35_ Aggregates Say ‘NO’ to Long-Term Potentiation in the Hippocampus through Activation of Stress-Induced Phosphatase 1 and Mitochondrial Na^+^/Ca^2+^ Exchanger

**DOI:** 10.3390/ijms231911848

**Published:** 2022-10-06

**Authors:** Alexander V. Maltsev, Anna B. Nikiforova, Natalia V. Bal, Pavel M. Balaban

**Affiliations:** 1Institute of Higher Nervous Activity and Neurophysiology, Russian Academy of Sciences, Butlerova 5A, 117485 Moscow, Russia; 2Institute of Theoretical and Experimental Biophysics, Russian Academy of Sciences, Institutskaya 3, Pushchino, 142290 Moscow, Russia

**Keywords:** amyloid peptides, Alzheimer’s disease, field excitatory postsynaptic potentials, nitric oxide, serine/threonine phosphatases, mitochondrial Na^+^/Ca^2+^ exchanger

## Abstract

The search for strategies for strengthening the synaptic efficiency in Aβ_25-35_-treated slices is a challenge for the compensation of amyloidosis-related pathologies. Here, we used the recording of field excitatory postsynaptic potentials (fEPSPs), nitric oxide (NO) imaging, measurements of serine/threonine protein phosphatase (STPP) activity, and the detection of the functional mitochondrial parameters in suspension of brain mitochondria to study the Aβ_25-35_-associated signaling in the hippocampus. Aβ_25-35_ aggregates shifted the kinase–phosphatase balance during the long-term potentiation (LTP) induction in the enhancement of STPP activity. The PP1/PP2A inhibitor, okadaic acid, but not the PP2B blocker, cyclosporin A, prevented Aβ_25-35_-dependent LTP suppression for both simultaneous and delayed enzyme blockade protocols. STPP activity in the Aβ_25-35_-treated slices was upregulated, which is reverted relative to the control values in the presence of PP1/PP2A but not in the presence of the PP2B blocker. A selective inhibitor of stress-induced PP1α, sephin1, but not of the PP2A blocker, cantharidin, is crucial for Aβ_25-35_-mediated LTP suppression prevention. A mitochondrial Na^+^/Ca^2+^ exchanger (mNCX) blocker, CGP37157, also attenuated the Aβ_25-35_-induced LTP decline. Aβ_25-35_ aggregates did not change the mitochondrial transmembrane potential or reactive oxygen species (ROS) production but affected the ion transport and Ca^2+^-dependent swelling of organelles. The staining of hippocampal slices with NO-sensitive fluorescence dye, DAF-FM, showed stimulation of the NO production in the Aβ_25-35_-pretreated slices at the dendrite-containing regions of CA1 and CA3, in the dentate gyrus (DG), and in the CA1/DG somata. NO scavenger, PTIO, or nNOS blockade by selective inhibitor 3Br-7NI partly restored the Aβ_25-35_-induced LTP decline. Thus, hippocampal NO production could be another marker for the impairment of synaptic plasticity in amyloidosis-related states, and kinase–phosphatase balance management could be a promising strategy for the compensation of Aβ_25-35_-driven deteriorations.

## 1. Introduction

Alzheimer’s disease (AD) is actually a severe challenge for basic and translational neuroscience, annually affecting millions of new patients and creating billions of dollars burden around the world [1,2]. Mechanistically, the progression of AD is correlated with the accumulation of misfolded structural proteins and their fragments in the brain, leading to neuroinflammation, neuronal loss, and, ultimately, a memory decline accompanied by cognitive dysfunctions [1,3]. The forming of amyloid fibrils, plaques, and agglomerates of misfolded tau protein in different models of AD pathology is well-documented, including post-mortem analysis of brains from people with AD [4,5,6]. Functionally, amyloid aggregates strongly suppress synaptic plasticity, leading to memory impairment in both invertebrate and vertebrate animal species, which apparently indicates there are conservative mechanisms mediating synaptic deteriorations [7,8,9]. One of the key amyloid peptides established in the respect of neuronal excitotoxicity is Aβ_25-35_, which has oligomers that disrupt the long-term potentiation (LTP) in crucial brain structures [10,11,12]. Unambiguous targets for Aβ_25-35_-driven signaling are still unknown, which is due to the complexity and prolongation of AD-developed events. Different studies indicate that amyloidosis could be accompanied by the downregulation of muscarinic [13,14], nicotinic cholinoreceptors’ sensitivity [2,15], AMPA receptor trafficking dysregulation [16,17], changes in the NMDA-receptor-mediated fluxes [18,19], upregulation of voltage-gated Ca^2+^ channels [20,21], or outward K^+^ current inhibition in neurons, or by its activation in microglia [22,23], the activation of ryanodine and inositol triphosphate receptors [24,25], or global mitochondrial bioenergetic disturbances [26,27]. Several studies showed the involvement of phosphatases in the development of a number of neurodegenerative models, including AD and parkinsonism. Thus, in mutant mice overexpressing the amyloid precursor protein (APP), an increase in the activity of PTP1B tyrosine phosphatase was found, which correlates with the development of neuroinflammation and the progression of cognitive deficit. Systemic blockade of PTP1B phosphatase partially prevented the deterioration of synaptic plasticity [28]. Another tyrosine phosphatase and tensin homolog (PTEN), which negatively regulates the prosurvival pathway PI3K-Akt-mTOR, increased the expression 48 h after Aβ_25-35_ application to organotypic cultures, which correlates with cell death [29]. A decrease in the activity of neuronal-specific striatal-enriched tyrosine phosphatase, STEP, prevented the internalization of the GluA1/GluA2 subunits of glutamate receptors, which is responsible for the partial suppression of the synaptic transmission induced by the action of amyloid aggregates [30]. Sparse and somewhat controversial data also exist for serine/threonine phosphatases. Thus, crosstalk between impaired phosphatase 2A (PP2A) activity and the cancerous inhibitor protein of PP2A (CIP2A) expression enhancement is responsible for the chronic increase in hyperphosphorylated tau protein [31]. At the same time, incubation of amyloid oligomers caused the induction of PP2A activity, since the decrease in LTP in amyloid-treated slices was prevented by the specific PP2A inhibitor foestriscin, as well as the application of pyruvate [32]. The inconsistency of data regarding the activity of serine/threonine phosphatases during amyloidosis progression indicates the complexity of amyloid-driven neurochemical events in the hippocampus, and, probably, at least in part, can be explained by the existence of critical ‘time windows’ in which the kinase–phosphatase balance can be bidirectionally switched.

In addition, the mechanisms underlying the pathogenesis of neurodegenerative states are closely associated with mitochondrial dysfunction and oxidative stress, an imbalance between the production and utilization of reactive oxygen species (ROS). According to the hypothesis of the ‘mitochondrial cascade’ [33,34], the decrease in the synthesis of adenosine triphosphate (ATP) and oxidative stress leads to an excessive synthesis of APP, which can directly have toxic effects on mitochondria, aggravating neurodegenerative processes [33,35]. Mitochondrial dysfunction results in suppressing the bioenergetics processes in nerve cells, the damage of membrane structures by free radicals, neuroinflammation, violation of the synaptic transmission, increase in the release of glutamate from the presynaptic terminals, decrease in the plasticity of synaptic contacts, and, ultimately, the death of neurons [35,36,37,38]. The ability of amyloid oligomers to initiate multiple intracellular signal pathways exacerbates the neurodegenerative processes. In the brains of patients with Aβ-amyloidosis, amyloids are accumulated in mitochondria and impair the activity of the glycolysis and tricarboxylic acid cycle enzymes, stimulating ROS production [39,40,41]. The direct action of oligomers and amyloid fibrils on the outer mitochondrial membrane (OMM) is a disruption of energy metabolism and electron chain transport [34,42]. Currently, the relationships between mitochondrial dysfunction and the kinase–phosphatase balance during amyloid-associated synaptic deteriorations is also studied quite fragmentarily. In this work, we used field excitatory postsynaptic potential (fEPSPs) recording, fluorescent NO-sensitive imaging in slices, direct measurements of serine/threonine phosphatase activity, and recordings of functional mitochondrial parameters (transmembrane mitochondrial potential and the ROS production and swelling of organelles in the suspension) to describe the Aβ_25-35_-driven neurochemical events in the hippocampus. We observed that Aβ_25-35_-dependent suppression of LTP is strictly related to the induction of PP1 activity and enhancement of NO production in CA1, CA3, and the dentate gyrus. Curiously, a delayed PP1/PP2A blockade 1 h after the Ab_25-35_ treatment of slices was quite effective in preventing fEPSP decline after LTP induction. The direct action of the Aβ_25-35_ aggregates on mitochondria revealed a moderate effect in the speed of mitochondrial swelling, an effect that disappeared during the increase in external Ca^2+^ in the chamber solution, which was well-correlated to the mitochondrial Na^+^/Ca^2+^ exchanger (mNCX) operations. Thus, the mNCX blockade strongly rescued the late LTP phase in Aβ_25-35_-treated slices. The data obtained clarified immediate influences of the Aβ_25-35_ oligomers on synaptic plasticity in the hippocampus and can be useful as a potential strategy for the compensation of some neurodegenerative states.

## 2. Results

### 2.1. Aβ_25-35_ Aggregates Impaired Synaptic Plasticity through PP1/PP2A-Sensitive Mechanism

To prepare Aβ_25-35_ aggregate solution, we used once-frozen 1 mM Aβ_25-35_ aliquots dissolved in mQ water. The final Aβ_25-35_ concentration in the ACSF was adjusted to 50 nM, after which the solution was kept for 24 h at +4 °C for monomer aggregation (Figure 1A).

Hippocampal slices after cutting and 1 h recovery were incubated at room temperature in the presence of the continuously oxygenized 50 nM Aβ_25-35_ aggregate solution for 1 h alone (Aβ_25-35_ group) or together with the STPP blockers, okadaic acid, or cyclosporin A, respectively (simultaneous enzyme blockade protocol, Figure 1A). Incubation of the hippocampal slices in the presence of Aβ_25-35_ aggregates strongly impaired the L-LTP phase, 3 h after the induction of tetanic stimulation (to 108.2 ± 7.4%, *n* = 9, from 204.8 ± 8.5% in control, *n* = 13, *p* ˂ 0.001, Figure 1B,C), without influence on the E-LTP phase (300.8 ± 30.4%, *n* = 9, vs. 311.3 ± 22.5%, *n* = 13, *p* = 0.95, Figure 1B,C). Moreover, Aβ_25-35_ aggregates suppressed the short-term plasticity associated with the release of neurotransmitters from presynaptic endings. For example, the paired pulse facilitation (PPF) ratio at the 50 ms ISI was decreased from 1.57 ± 0.06 in the control to 1.33 ± 0.05 in the Aβ_25-35_-treated slices for the pre-tetanic state (*n* = 9, *p* = 0.006, Figure 1D) and from 1.55 ± 0.07 in control to 1.32 ± 0.07 in the Aβ_25-35_-treated slices for the L-LTP, 3 h after tetanus induction (*n* = 9, *p* = 0.008, Figure 1D). Furthermore, the difference in PPF ratios in the Aβ_25-35_-treated slices vs. the control was statistically significant at the 30, 50, and 100 ms ISIs, for both pre- and post-tetanic states (Figure 1E). For example, at the 100 ms ISI, the PPF ratios were 1.27 ± 0.05 vs. 1.42 ± 0.04 (*n* = 9, *p* = 0.038, Figure 1E, left panel) and 1.25 ± 0.04 vs. 1.43 ± 0.03 (*n* = 9, *p* = 0.004, Figure 1E, right panel) for pre-tetanus and 3 h after LTP induction, respectively. The decrease in PPF ratios in the Aβ_25-35_-treated slices could reflect the (in)direct influences of the Aβ_25-35_ aggregates on the presynapse. Coincubation of slices in the presence of the Aβ_25-35_ and PP1/PP2A inhibitor, okadaic acid (OA, 100 nM), restored the L-LTP phase, 3 h after tetanic induction (to 170.1 ± 10.5%, *n* = 9, *p* = 0.077 vs. control, *p* ˂ 0.001 vs. Aβ_25-35_ group, Figure 1B,C). Curiously, PPF at the 50 ms ISI in the Aβ_25-35_ + OA-treated slices during LTP had a tendency to increase in comparison to the control (40 min after tetanus: 1.68 ± 0.07 vs. 1.48 ± 0.05 in control, *n* = 9, *p* = 0.15; 2 h after tetanus: 1.86 ± 0.07 vs. 1.61 ± 0.06 in control, *n* = 9, *p* = 0.07, Figure 1D, Appendix A), which was statistically significant for the L-LTP, 3 h after tetanus (2.22 ± 0.12 vs. 1.66 ± 0.08 in control, *n* = 9, *p* ˂ 0.001, Figure 1D, Appendix A). At the same time, PPF analysis revealed that the OA did not change the impaired PPF ratios before tetanic induction (at the 50 ms ISI: 1.32 ± 0.06 vs. 1.34 ± 0.05 in the Aβ_25-35_-treated slices, *n* = 9, *p* = 0.93, Figure 1D,E). The OA alone (100 nM) also significantly increased the PPF ratios at the 30, 50, 100 ms ISIs in both the pre- and post-tetanic states. For example, at 50 ms ISI, the PPF ratios were 1.91 ± 0.10 (*p* = 0.003 vs. control) and 1.90 ± 0.12 (*p* = 0.028 vs. control) for pre-tetanus and 3 h after LTP induction, respectively (for both *n* = 9, Figure 1D,E). Thus, amyloid-aggregate-induced L-LTP suppression is, apparently, related to the PP1/PP2A-driven signaling pathway(s).

### 2.2. PP2B Is Not Involved in the Aβ_25-35_-Dependent Deteriorations of Synaptic Plasticity

Further, we used an inhibitor of PP2B, cyclosporin A (CsA, 5 μM), to determine the putative role of this STPP member for the Aβ_25-35_-mediated neurochemical events in the hippocampus. CsA alone (5 μM) did not influence the E-LTP phase (297.9 ± 20.5, *n* = 9, vs. 300.8 ± 30.4%, *p* = 0.99, Figure 2A,B) nor the L-LTP phase (202.5 ± 9.9%, *n* = 9, vs. 204.8 ± 8.5% in control, *p* = 0.68, Figure 2A,B). Coincubation of the hippocampal slices in the Aβ_25-35_ + CsA did not restore Aβ_25-35_-induced L-LTP impairment (108.9 ± 8.8%, *n* = 9, vs. 108.2 ± 7.4% in the Aβ_25-35_-treated group, *p* = 0.73, Figure 2A,B). Furthermore, CsA (5 μM) did not prevent a decrease in the PPF ratios in the Aβ_25-35_ + CsA-treated slices at the 30, 50, and 100 ms ISIs for both pre- and post-tetanic states (Figure 2C,D).

For example, the before-LTP-induction (pre-tet) PPF ratios were 1.27 ± 0.03 vs. 1.34 ± 0.04 in Aβ_25-35_-treated slices (*p* = 0.12) and 1.23 ± 0.02 vs. 1.33 ± 0.05 in Aβ_25-35_-treated slices (*p* = 0.07) for 30 and 50 ms ISIs, respectively (for all *n* = 9, Figure 2C,D, left panel). PPF values for 30 and 50 ms ISIs in the L-LTP of the CsA+Aβ_25-35_ group, 3 h after tetanus, were, respectively, 1.27 ± 0.03 vs. 1.32 ± 0.07 in Aβ_25-35_-treated slices (*p* = 0.79) and 1.25 ± 0.02 vs. 1.31 ± 0.05 in Aβ_25-35_-treated slices, *p* = 0.38 (for all *n* = 9, Figure 2C,D, right panel). Thus, PP2B is, apparently, not involved in the Aβ_25-35_-mediated signaling processes in the CA3–CA1 hippocampal synapses.

### 2.3. Direct Measurements of STPP Activity in Hippocampal Slices

Using the protocol described in the Materials and Methods section, we colorimetrically estimated STPP activity in the control, tetanized slices at different times after tetanus induction: 40 min for E-LTP, 3 h after tetanus for L-LTP, slices pretreated in Aβ_25-35_ for 1 h or 3 h of incubation, as well as tetanized Aβ_25-35_-treated slices in the E-LTP and L-LTP phases. The specificity of phosphatase reaction was tested by STPP inhibitors, OA (100 nM) or CsA (5 μM), respectively. Figure 2E, left panel, indicates the spectra of optical density for the control, Aβ_25-35_-treated samples, as well as for Aβ_25-35_+OA and Aβ_25-35_ + CsA-treated samples, with maximal absorbance at 640 nm. Normalized to the total protein amounts, activities are presented in diagrams. During E-LTP, the STPP activity was reduced to 77.6 ± 5.1% from the control values (*n* = 4, *p* = 0.029, Figure 2E, right panel), denoting a shift in the kinase–phosphatase balance, to the enhancement of the protein kinase’s activity occurring after tetanic stimulation. Curiously, significant increases of STTP activity were observed in Aβ_25-35_-treated slices for E-LTP (135.5 ± 10.5%, *n* = 4, *p* ˂ 0.001 vs. both control and E-LTP values) as well as for L-LTP (128.1 ± 5.2%, *n* = 4, *p* = 0.018 vs. control, *p* = 0.057 vs. L-LTP values, Figure 2E, right panels). Incubation of hippocampal slices in Aβ_25-35_ for 1 h or 3 h also significantly increased the STPP activity to 149.8 ± 6.3% (*n* = 4, *p* ˂ 0.001 vs. control) and 148.7 ± 8.4% (*n* = 4, *p* ˂ 0.001 vs. control, Figure 2E, right panels), respectively. The OA (100 nM) in all cases fully prevented the Aβ_25-35_-dependent increase in STPP activity to 96.8 ± 3.8% (in the Aβ_25-35_ + OA group, *n* = 4, *p* ˂ 0.001 vs. the Aβ_25-35_-treated group), to 91.4 ± 3.0% (in the Aβ_25-35_ + OA-E-LTP group, *n* = 4, *p* = 0.002 vs. Aβ_25-35_-E-LTP), and to 97.3 ± 4.5% (in the Aβ_25-35_ + OA-L-LTP group, *n* = 4, *p* = 0.03 vs. Aβ_25-35_-L-LTP, Figure 2E, right panels). At the same time, CsA (5 µM) actually did not prevent the Aβ_25-35_-mediated increases in STPP activities, in comparison to the Aβ_25-35_, Aβ_25-35_-E-LTP, and Aβ_25-35_-L-LTP groups (for all *n* = 4, *p* > 0.05, Figure 2E, right levels). Thus, STPP activity measurements confirmed the PP1/PP2A-sensitive mechanisms for Aβ_25-35_ influences on the synaptic transmission in the hippocampus.

### 2.4. Delayed Enzyme Blockade Protocol for STPP Influence on Synaptic Transmission

To study the possibility of compensation for synaptic deteriorations in Aβ_25-35_-treated slices, we used another protocol, a delayed enzyme blockade (Figure 3A). In these experiments, an STPP blocker (OA or CsA) was applied to the chamber solution after 1 h incubation of slices in the Aβ_25-35_ aggregates, 20 min before tetanic induction and for 2 h during LTP, after which it was washed. The OA (100 nM) partially prevented Aβ_25-35_-induced L-LTP suppression to 159.4 ± 8.0% vs. 108.2 ± 7.4% in the Aβ_25-35_-treated group (*n* = 9, *p* = 0.01 vs. control, *p* = 0.003 vs. Aβ_25-35_-treated group, Figure 3B,C). As in the case of the protocol for the simultaneous enzyme blockade, we observed a restoration of the decreased PPF ratios at the 30–100 ms ISIs in the Aβ_25-35_ + OA group, toward the end of the L-LTP phase, 3 h after tetanic stimulation (Figure 3D,E). For example, at the 30 and 50 ms ISIs, the PPF ratios were, respectively, 2.03 ± 0.10 vs. 1.32 ± 0.07 in the Aβ_25-35_-treated slices (*p* < 0.001) and 1.95 ± 0.08 vs. 1.31 ± 0.05 in the Aβ_25-35_-treated slices, *p* < 0.001 (for all *n* = 9, Figure 3D,E). The CsA (5 μM) in the protocol of the delayed enzyme blockade did not affect the Aβ_25-35_-induced LTP suppression as expected (Appendix A). For the L-LTP phase, the fEPSP decrease in the Aβ_25-35_ + CsA group was 110.5 ± 7.3% (*n* = 9, *p* = 0.62 vs. Aβ_25-35_-treated, Appendix A). Thus, the data obtained suggest that the disturbance in the synaptic transmission after the Aβ_25-35_ aggregate’s influences on the CA3–CA1 synapses can, at least partly, be rescued in vitro with a delay.

### 2.5. Which Phosphatase(s) Is Involved in OA-Associated Attenuation of Aβ_25-35_-Induced LTP Suppression?

Next, to answer the question, which phosphatase(s) is responsible for the Aβ_25-35_-mediated LTP suppression, we used two selective inhibitors, sephin1 (Se1), which blocks the stress-induced isoform PP1α, and cantharidin (Cthr), which ceases PP2A activity. Se1 (10 μM) alone had no influence on the E-LTP (278.2 ± 24.4%, *n* = 9, *p* = 0.46 vs. control, Figure 4A top panel, Figure 4B,C), the L-LTP (190.3 ± 14.3%, *n* = 9, *p* = 0.34 vs. control, Figure 4A–C) or the PPF ratios (Figure 4D,E). For example, at the 50 ms ISI, the PPF values were 1.62 ± 0.04 (*n* = 9, *p* = 0.32 vs. control, Figure 4A bottom panel, Figure 4D) and 1.65 ± 0.05 (*n* = 9, *p* = 0.18 vs. control, Figure 4A bottom panel, Figure 4E) for pre-tetanic levels and post-tetanus, 3 h after LTP induction, respectively. At the same time, the PP1α blockade by Se1 during Aβ_25-35_ aggregate slice incubation prevented Aβ_25-35_-mediated L-LTP suppression, to 182.9 ± 10.2% (*n* = 9, *p* = 0.065 vs. control, *p* ˂ 0.001 vs. Aβ_25-35_-treated group, Figure 4A top panel, Figure 4B,C). Moreover, Se1 restored PPF ratios to control levels at the 30–100 ms ISIs; for example, the PPF ratios in post-tetanic recordings, 3 h after LTP induction, were 1.70 ± 0.06 (*n* = 9, *p* = 0.003 vs. Aβ_25-35_-treated group, Figure 4E) and 1.64 ± 0.05 (*n* = 9, *p* = 0.002 vs. Aβ_25-35_-treated group, Figure 4E) for 30 and 50 ms, respectively.

In contrast, Cthr (10 μM) alone impaired the L-LTP phase (142.2 ± 12.9%, *n* = 9, *p* ˂ 0.001 vs. control, Figure 4F top panel, Figure 4G,H), without influencing the E-LTP phase (267.2 ± 16.4%, *n* = 9, *p* = 0.42 vs. control, Figure 4F top panel, Figure 4G,H). Cthr increased the PPF ratios at all studied intervals (30–400 ms), so the PPF ratios were 1.82 ± 0.06 (*n* = 9, *p* = 0.008 vs. control, Figure 4F bottom panel, Figure 4I) and 1.99 ± 0.10 (*n* = 9, *p* = 0.005 vs. control, Figure 4F bottom panel, Figure 4J) for the pre-tetanic level and at the end of L-LTP, respectively. However, Cthr was ineffective for rescuing Aβ_25-35_-dependent L-LTP suppression (95.3 ± 10.1%, *n* = 9, *p* ˂ 0.001 vs. control, *p* = 0.4 vs. Aβ_25-35_-treated group, Figure 4F top panel, Figure 4G,H). Thus, the obtained data suggest that PP1α is involved in the Aβ_25-35_ effects, rather than PP2A.

### 2.6. Mitochondria and Aβ_25-35_ Aggregates

Amyloid-driven stress signaling also affects the exacerbation of mitochondrial bioenergetics, dysregulation of mitochondrial ion transport, and changes in ATP synthesis and phosphorylation. We used an inhibitor of the mitochondrial Na^+^/Ca^2+^ exchanger (mNCX), CGP37157, to address the question of the participation of this ion transporter in Aβ_25-35_-dependent L-LTP suppression. CGP37157 (10 μM) alone had no influence on the E-LTP phase (279.1 ± 22.6%, *n* = 9, *p* = 0.55 vs. control, Figure 5A top panel, Figure 5B,C), the L-LTP phase (193.5 ± 9.4%, *n* = 9, *p* = 0.17 vs. control, Figure 5A, top panel, Figure 5B,C), or short-term plasticity (Figure 5A bottom panel, Figure 5D,E).

For example, at the 50 ms ISI, the PPF values were 1.59 ± 0.05 (*n* = 9, *p* = 0.26 vs. control, Figure 5A bottom panel, Figure 5D) and 1.63 ± 0.04 (*n* = 9, *p* = 0.23 vs. control, Figure 5A bottom panel, Figure 5E) for pre-tetanic levels and 3 h after LTP induction, respectively. At the same time, the mNCX blockade by CGP37157 during Aβ_25-35_ aggregate slice incubation prevented Aβ_25-35_-mediated L-LTP suppression, to 183.1 ± 13.4% (*n* = 9, *p* = 0.14 vs. control, *p* ˂ 0.001 vs. Aβ_25-35_-treated group, Figure 5A top panel, Figure 5B,C). Moreover, CGP37157 restored the PPF ratios in the Aβ_25-35_-treated slices to control levels, for the 30–100 ms ISIs. For example, the PPF ratios in post-tetanic recordings, 3 h after LTP induction, were 1.67 ± 0.08 (*n* = 9, *p* = 0.017 vs. Aβ_25-35_-treated group, Figure 5E) and 1.57 ± 0.05 (*n* = 9, *p* = 0.042 vs. Aβ_25-35_-treated group, Figure 5A bottom panel, Figure 5E) for 30 and 50 ms, respectively. Thus, mNCX is, apparently, another promising target for the compensation of Aβ_25-35_-associated synaptic insufficiency, at least in the CA3–CA1 synapses.

Furthermore, we directly evaluated the Aβ_25-35_ effect on the mitochondrial functionality in the suspension of brain mitochondria. Aβ_25-35_ (10–200 nM), practically, did not change the transmembrane mitochondrial potential Δ*Ψ_m_* detected by fluorescence dye rhodamine 123 (Figure 5F). The ratio of rhodamine 123 fluorescence in Aβ_25-35_-treated mitochondria to the rhodamine 123 fluorescence in the presence of 200 nM FCCP, an established uncoupler of oxidative phosphorylation, was used to estimate the influence of Aβ_25-35_ on the Δ*Ψ_m_*. The closer this value is to 1, the more the Δ*Ψ_m_* potential is reset. We did not observe any statistically significant differences in all probes; however, in the conditions of 5 mM pyruvate + 5 mM malate as substrates, 10 nM Aβ_25-35_ had a tendency to uncouple and dissipate the Δ*Ψ_m_* (0.94 ± 0.06 vs. 0.88 ± 0.06 in control, *p* = 0.082, *n* = 3, Figure 5F). Similar data were obtained in the presence of succinate 5 mM + rotenone 1 μM as substrates (Appendix A). Moreover, Aβ_25-35_ (10–200 nM) did not change the production of superoxide O_2_^−•^ assessed by the chemiluminescence dye, MCLA (Figure 5G). Superoxide dismutase (SOD) as the negative control decreased MCLA luminescence (from 13284 ± 94 a.u. to 6366 ± 73 a.u., *p* ˂ 0.001, Figure 5G), whereas MCLA luminescence levels in the Aβ_25-35_ groups were not different from those of the control samples (for example, for 10 nM Aβ_25-35_—13387 ± 65 a.u., *p* = 0.96 vs. control, Figure 5G). Interestingly, we detected an Aβ_25-35_ influence on the Ca^2+^-dependent swelling of mitochondria, a process which is closely related to the global changes in mitochondrial ion transport. Mitochondrial swelling was detected as a decrease in the optical density absorption spectra at 540 nm in different conditions: with the addition of 50 μM Ca^2+^ to the organelle suspension (Figure 5H right panels) or without it (Figure 5H, left panels). An Aβ_25-35_ (10–200 nM) dose independently protected mitochondria from the swelling in the Ca^2+^-free medium and had no protection effect in the presence of 50 μM Ca^2+^. So, the optical densities in 10, 50, 100, 150, and 200 nM Aβ_25-35_-treated mitochondria without external Ca^2+^ were 0.718 ± 0.014, 0.720 ± 0.011, 0.707 ± 0.015, 0.723 ± 0.019, and 0.722 ± 0.010 a.u., respectively (vs. 0.563 ± 0.010 a.u in control probes, for all *n* = 3, *p* ˂ 0.05, Figure 5H, left panels). Dose-independent protection against mitochondrial swelling disappeared in the presence of 50 µM Ca^2+^ (for example, optical density in the 10 nM Aβ_25-35_ was 0.371 ± 0.024 a.u. vs. 0.346 ± 0.026 a.u. in control, *n* = 3, *p* = 0.7, Figure 5H, right panels). These observations are well consistent with the mNCX mode operations. CGP37157-sensitive mNCX operation is responsible for the inverted ion antiport, increasing the Ca^2+^ leakage from mitochondria into the cytosol. The mNCX-dependent Ca^2+^ output somewhat increases mitochondrial Ca^2+^ capacity, attenuating the swelling of mitochondria. In the conditions of increased external Ca^2+^, the mNCX drives the forward mode, loading Ca^2+^ into mitochondria together with Na^+^ outflow to the cytosol. Thus, inversion of the ion antiport for Ca^2+^ and Na^+^ attenuated the mNCX-dependent protection effect against mitochondrial swelling.

### 2.7. Aβ_25-35_ Aggregates ‘Say’ NO to the LTP in CA3–CA1 Synapses

Previously, we showed that LTP suppression in several neuropathological conditions such as protein synthesis inhibitors’ application was accompanied by an increase in the NO production, suggesting the potential role of NO in labeling some neurophysiological stress reactions [43,44]. In the next series of experiments, a direct estimation of NO production in the hippocampus was completed. In non-stimulated hippocampal slices, the treatment by 50 nM Aβ_25-35_ significantly increased the NO-sensitive fluorescence of the DAF-FM probe in key hippocampal layers including CA1 somas (*Stratum pyramidale*), CA1 dendrites (*Stratum radiatum*, SR; and *Stratum lacunosum-moleculare*, SLM), CA3 dendrites, dentate gyrus (DG) dendrites, and somas (Figure 6).

In the CA1 dendrites (SLM), the Aβ_25-35_-dependent NO synthesis increased to 134.4 ± 3.3% (*n* = 9, *p* ˂ 0.001, Figure 7, Appendix A). In the CA3 dendrites, the NO production increase was 137.3 ± 2.6% (*n* = 9, *p* ˂ 0.001, Figure 7, Appendix A). In the DG dendrites, the NO synthesis heightened to 162.6 ± 3.2% (*n* = 9, *p* < 0.001, Figure 7, Appendix A). Moreover, Aβ_25-35_ strongly increased NO production in the CA1 somas (SP, to 136.4 ± 2.5%, *n* = 9, *p* < 0.001, Figure 7, Appendix A) and DG somas (to 165.3 ± 3.9%, *n* = 9, *p* < 0.001, Figure 7, Appendix A), without a significant increase in the CA3 somas (104.1 ± 1.4%, *n* = 9, *p* > 0.05, Figure 6 and Figure 7). The NO synthesis in tetanized Aβ_25-35_-pretreated slices was potentiated in comparison to the non-stimulated Aβ_25-35_-treated slices in the SLM (153.5 ± 4.9%, *n* = 9, *p* = 0.01, Figure 7) and CA3 dendrites (162.7 ± 3.9%, *n* = 9, *p* = 0.001, Figure 7) but not for those in DG (164.1 ± 4.2%, *n* = 9, *p* = 0.93, Figure 7).

Interestingly, OA (100 nM) canceled the Aβ_25-35_-driven DAF-FM fluorescence’s increase in SP (105.1 ± 2.0%, *n* = 9, Figure 7) and partly limited the NO production in other regions, such as SLM (to 116.6 ± 2.4%, *n* = 9, *p* = 0.002 vs. Aβ_25-35_ group, Figure 7), CA3 dendrites (117.0 ± 2.5%, *n* = 9, *p* = 0.001 vs. Aβ_25-35_ group, Figure 7), DG somas (135.8 ± 4.2%, *n* = 9, *p* = 0.001 vs. Aβ_25-35_ group, Figure 7), and dendrites (134.0 ± 4.4%, *n* = 9, *p* = 0.001, vs. Aβ_25-35_ group, Figure 7). The same effects of the OA were observed in tetanized Aβ_25-35_-treated slices, confirming at least partial involvement for PP1/PP2A in the Aβ_25-35_-dependent NO synthesis in the hippocampus. In contrast, the blockade of PP2B by CsA (5 µM) had no influence on the Aβ_25-35_-dependent NO production in non-stimulated or in tetanized slices (Figure 7). Taken together, the data obtained suggest that PP1/PP2A mediates the Aβ_25-35_-induced NO production, which can be one of the regulators for the L-LTP in the CA3–CA1 synapses.

To disrupt the NO signaling, we used a NO scavenger, PTIO (50 µM), and a neuronal NO synthase blocker, 3Br-7NI (5 µM). PTIO (50 μM) alone had no influence on the E-LTP (268.9 ± 20.0%, *n* = 9, *p* = 0.42 vs. control, Figure 8A top panel, Figure 8B,C) or short-term plasticity (Figure 8D,E) but impaired the L-LTP (161.3 ± 8.7%, *n* = 9, *p* = 0.014 vs. control, Figure 8A top panel, Figure 8B,C).

At the 50 ms ISI, the PPF values were 1.50 ± 0.07 (*n* = 9, *p* = 0.46 vs. control, Figure 8A bottom panel, Figure 8D) and 1.49 ± 0.05 (*n* = 9, *p* = 0.5 vs. control, Figure 8A bottom panel, Figure 8E) for pre-tetanic levels and post-tetanus, 3 h after LTP induction, respectively. At the same time, NO scavenging by PTIO during Aβ_25-35_ aggregate slice incubation partly diminished the Aβ_25-35_-mediated L-LTP suppression to 167.3 ± 10.4% (*n* = 9, *p* = 0.04 vs. control, *p* = 0.001 vs. Aβ_25-35_-treated group, Figure 8A top panel, Figure 8B,C). Moreover, PTIO restored the PPF ratios to control levels at the 30–100 ms ISIs. For example, PPF ratios in post-tetanic recordings, 3 h after LTP induction, were 1.62 ± 0.07 (*n* = 9, *p* = 0.01 vs. Aβ_25-35_-treated group, Figure 8E) and 1.55 ± 0.06 (*n* = 9, *p* = 0.017 vs. Aβ_25-35_-treated group, Figure 8A bottom panel, Figure 8E) for 30 and 50 ms, respectively. Similarly, 3Br-7NI (5 μM) alone impaired the L-LTP phase (168.3 ± 7.1%, *n* = 9, *p* = 0.009 vs. control, Figure 8F top panel, Figure 8G,H), without influence on the E-LTP phase (295.7 ± 10.1%, *n* = 9, *p* = 0.89 vs. control, Figure 8F top panel, Figure 8G,H). Moreover, 3Br-7NI did not change the PPF ratios at all studied intervals (30–400 ms); for example, at the 50 ms ISI, the PPF ratios were 1.54 ± 0.04 (*n* = 9, *p* = 0.95 vs. control, Figure 8F bottom panel, Figure 8I) and 1.66 ± 0.08 (*n* = 9, *p* = 0.51 vs. control, Figure 8F bottom panel, Figure 8J) for pre-tetanic level and at the end of L-LTP, respectively. The 3Br-7NI was partly effective in the prevention of Aβ_25-35_-dependent L-LTP suppression (157.8 ± 5.3%, *n* = 9, *p* = 0.001 vs. control, *p* = 0.001 vs. Aβ_25-35_-treated group, Figure 8F top panel, Figure 8G,H). Thus, it can be assumed that NO in the hippocampus is closely connected with the Aβ_25-35_-mediated synaptic transmission inhibition, and the breakdown of NO signaling, at least in vitro, could be useful for the preservation of fEPSP responses in the CA3–CA1 synapses.

## 3. Discussion

Amyloid peptide aggregates noticeably change ion channel transport, leading to adaptive neurochemical stress responses in the neuronal and glial cells [45,46,47]. At the synaptic level, Aβ oligomers suppress post-tetanic and long-term potentiation, decreasing the number of released vesicles [48,49,50]. In accordance with these data, we revealed the decreased PPF ratio at 30–100 ms interstimulus intervals in the Aβ_25-35_-treated hippocampal slices (Figure 1E). Furthermore, 1 h incubation of slices in the presence of 50 nM Aβ_25-35_ disrupted the late LTP phase, 3 h after the tetanic stimulation (Figure 1A,B), which was previously shown for the different stimulations’ protocols of invertebrates and mammal species [7,51,52,53]. Curiously, the Aβ_25-35_ monomers in the freshly prepared amyloid solution (50 nM) did not change the LTP kinetics, after tetanic stimulation of the Aβ_25-35_-pretreated hippocampal slices (Appendix A). Impairment of both short-term and long-term plasticity denotes the multiple influences of the Aβ_25-35_ aggregates at pre- and postsynaptic endings. In this paper, we observed at least three important players: the PPPR15A subunit of the PP1 phosphatase complex (PP1α), the mitochondrial Na^+^/Ca^2+^ exchanger (mNCX), and the neuronal NO synthase (nNOS) involved in the Aβ_25-35_-associated neurochemical events in the rat hippocampus.

The regulatory subunit of the protein phosphatase 1 complex (PPPR15A), well known as GADD34 (growth arrest and DNA damage-inducible gene), is a cell stress protein that recruits PP1α activity under conditions of oxidative, genotoxic, and other forms of stress [54,55]. Immunocytochemistry analysis revealed GADD34 localization both in the cytosol and membrane organelles, such as the endoplasmic reticulum, Golgi apparatus, and mitochondria [56]. GADD34 trafficking between cytosol and the intracellular compartments drives its proteasomal degradation, changing the dephosphorylation status of PP1-targeted proteins [56,57]. In our experiments, the specific blocker of GADD34, sephin1, strongly prevented Aβ_25-35_-dependent LTP suppression in the protocols of both the transient and chronic blockade of the enzyme (Figure 4). Curiously, sephin1 regulates integrated stress response (ISR), preventing the accumulation of misfolded proteins and promising therapeutic potency against neuronal excitotoxicity, multiple sclerosis, and neurodegeneration [58,59]. Moreover, GADD34-expression enhancement is a hallmark of some neuropathological states, including traumatic brain injury, cerebral ischemia/reperfusion injury, and glioblastoma [60,61,62]. Furthermore, increased GADD34 activity was observed in glial cells, such as oligodendrocytes during AD pathology development [63], and overexpression was observed in AD transgenic J20 mice [64].

Direct measurements of serine/threonine phosphatase (STPP) activity revealed that the incubation of hippocampal slices in the presence of Aβ_25-35_ amyloid aggregates significantly elevates it (Figure 2E). An Aβ_25-35_-mediated STPP activity increase was prevented by the PP1/PP2A blockade by okadaic acid but not by the PP2B blocker cyclosporin A (Figure 2E). Considering the fact that a selective PP2A inhibitor, cantharidin, was ineffective in preventing Aβ_25-35_-dependent LTP suppression (Figure 4F,H), the STTP increase is most likely related with the induction of PP1 activity rather than that of PP2A. It should be noted that the application of Aβ_25-35_ aggregates switches the kinase–phosphatase balance to a prevalence of phosphatase activity (Figure 2E), whereas during the E-LTP phase in control slices STPP activity was strongly downregulated (Figure 2E). Probably, STPP-activity enhancement is partly involved in the impairment of synaptic transmission during LTP development in the Aβ_25-35_-pretreated slices. During the E-LTP phase, the stimulation of multiple protein kinases, including serine/threonine A, B, C, and G and Ca^2+^-calmodulin-dependent II occurs, which is crucial for the enhancement of synaptic transmission in tetanized slices [65,66,67]. The induction of phosphatase activity during the E-LTP phase can interfere the kinase-mediated signaling, leading to the disappearance of fEPSPs and the impairment of synaptic effectivity.

Earlier, we observed that the increase in PP2B activity during LTP suppression by protein synthesis inhibitors could be associated with the increase in nitric oxide (NO) production by neuronal NO synthase [68]. Thus, NO is a neuromediator, which supports synaptic plasticity and is involved in the reconsolidation and memory storage processes [69,70,71,72], and it can be a hallmark of synaptic deteriorations. NO-sensitive imaging of hippocampal slices reveals that Aβ_25-35_ aggregates strongly increase the NO levels in both the somas and dendrites of the CA1 and DG layers, as well as in the CA3 inputs including Shaffer’s collaterals (Figure 6 and Figure 7). The PP2B inhibitor cyclosporin A was ineffective in preventing Aβ_25-35_-dependent NO synthesis (Figure 7). In contrast, a PP1/PP2A blocker, okadaic acid, partly prevented the Aβ_25-35_-mediated NO increase in the CA1 and DG somas, as well as in the CA1, CA3, and DG dendrites (Figure 7). The enhancement of neuronal NOS activity can be associated with the PP1-dependent changes in the phosphorylation status of NOS. Thus, the nNOS dephosphorylation, when ‘inhibiting’ Ser-847 (Ser-852 in mice) or Thr-1296 residues by the PP1 complex, increases the NO production in different cell models [73,74,75,76]. Furthermore, PP1 could indirectly lead to the enhancement of nNOS phosphorylation when ‘stimulating’ Ser-1417 (Ser-1412 in mice) [77,78,79], which is also well-correlated with the increase in NO production in the Aβ_25-35_-pretreated hippocampal slices.

The involvement of mitochondria in the development of neurodegenerative states, including AD, is well-known as the concept ‘mitochondrial cascade’ [33,34]. The interaction between oxidative stress and mitochondrial dysfunction and the impairment of Ca^2+^ homeostasis induced by misfolded tau and β-amyloid play important roles in the progressive neuronal loss occurring in specific areas of the brain [80,81]. In our experiments, Aβ_25-35_ aggregate mediated LTP suppression, which was partly prevented in the presence of CGP37187, a blocker of the reverse mode of the mitochondrial Na^+^/Ca^2+^ exchanger (mNCX) (Figure 5A,C). The mNCX represents an ion transporter in the inner mitochondrial membrane, which carries three Na^+^ outwards in exchange for the one Ca^2+^ that is uploaded into the mitochondrion [82,83]. The reverse mode of the mNCX is responsible for the inverted ion antiport, increasing Ca^2+^ leakage from mitochondria into the cytosol [84,85]. The heightened [Ca^2+^]_in_ could lead to the additional elevation of NO synthase activity, through Ca^2+^-calmodulin protein–protein interaction [86], and it can be involved in the regulation of synaptic transmission, through exocytosis of the neuromediator vesicles [87,88]. Apparently, incubation of hippocampal slices in the presence of Aβ_25-35_ amyloid aggregates depletes the intracellular neuromediator sources, which is well-correlated with the significant decrease in the PPF ratio in the CA3–CA1 synapses (Figure 1E). Although CGP37187 did not alter the short-term plasticity and had no influence on the LTP kinetics (Figure 5A,C), the data obtained strongly indicate mNCX’s participation in the Aβ_25-35_ amyloid response in the rat hippocampus. At the same time, neuronal deletion of the mitochondrial Na^+^/Ca^2+^ exchanger (NCLX, Slc8b1 gene) accelerated memory decline and increased amyloidosis, tau pathology, and, ultimately, memory loss in 3XTg-AD mice, [89]. Inhibition of the mNCX protects striatal neurons from α-synuclein cytotoxicity in models of Parkinson’s disease [90]. In addition, NCX1, NCX2, and NCX3 isoforms were upregulated in the synaptic terminals, accumulating amyloid-beta (Aβ), the neurotoxic peptide responsible for AD neurodegeneration [91,92]. More recently, the hyperfunction of a specific NCX subtype, NCX3, has been shown to delay endoplasmic reticulum stress and apoptotic neuronal death in hippocampal neurons exposed to Aβ insult [93]. Direct measurements of the mitochondrial parameters indicated that Aβ_25-35_ could directly regulate some mitochondrial functions, such as swelling and Ca^2+^ uptake (Figure 5H). Aβ_25-35_, in nanomolar concentrations, did not change the transmembrane mitochondrial potential (Figure 5F) or ROS production (Figure 5G). Curiously, the Aβ_25-35_ effects, in respect to the swelling of brain mitochondria, disappeared with an increase in the external Ca^2+^ in the chamber solution (Figure 5H, right panels). This fact is in good agreement with the operation of the mNCX modes. At heightened external Ca^2+^ concentrations, Ca^2+^ transport from mitochondria through the mNCX is not working [85,94]. The insufficient influence of mNCX-mediated Ca^2+^ fluxes, in respect to mitochondrial transmembrane potential, could be explained by the involvement in ΔΨ_m_ maintenance of major players, such as the mitochondrial Ca^2+^ uniporter (MCU), which operates a large Ca^2+^ influx to mitochondria, successfully compensating for the outwards mNCX-dependent Ca^2+^ leak.

At a minimum, the Aβ_25-35_-driven PP1α- and mNCX-related events may occur independently, because purified mitochondrial suspension, apparently, contains low quantities of GADD34 protein. However, there are indirect data concerning the mNCX’s modulation by mitochondrial serine/threonine PTEN-induced kinase 1 (PINK1) [95]. In this case, it is logical to suggest the participation of PP1α as another signaling player that provides negative feedback for the kinase–phosphatase switching for the mNCX’s transport regulation. On the other hand, changes in the intracellular [Ca^2+^]_in_, as in the case of the mNCX’s reverse mode stimulation, could underlie the enhancement of PP1α activity. Thus, endoplasmic reticulum store depletion (ER stress) is sufficient to induce GADD34 mRNA expression, which can be the mechanism, for example, of the calcium homeostasis disturbances during transient forebrain ischemia in rats [96,97]. Thus, a positive correlation between PP1α and the mNCX blockade was observed, in respect to the prevention of the Aβ_25-35_ aggregate-dependent LTP suppression (Appendix A).

Figure 9 presents a speculative scheme of Aβ_25-35_-driven signaling in the hippocampal neurons. Aβ_25-35_ aggregates (in)directly induced GADD34 and PP1α phosphatase complex activation. PP1α can dephosphorylate nNOS, leading to the disinhibition of nNOS activity followed by NO production. NO acts both pre- and postsynaptically, reducing the number of mediator vesicles and nitrosylating the protein targets, such as ion channels, enzymes, etc. Furthermore, Aβ_25-35_ aggregates stimulate the mNCX’s transport, elevating [Ca^2+^]_in_ in the cytosol. An increase in [Ca^2+^]_in_ can, additionally, facilitate nNOS activity through the Ca^2+^-calmodulin binding site at the nNOS. Moreover, the elevation of [Ca^2+^]_in_ can provide additional GADD34 induction and the enhancement of PP1α phosphatase activity. PP1α in the postsynaptic cells interferes with kinase-mediated signals, preventing fEPSP maintenance after LTP induction. PP1α in the presynapse may limit vesicular exocytosis of the neuromediator, impairing short-term plasticity.

## 4. Materials and Methods

### 4.1. Hippocampal Slice Preparation

All experimental procedures were conducted in accordance with the European Communities Council Directive of 24 November 1986 (86/609/EEC), on the protection of animals used for scientific purposes. The study protocol was approved by the Ethics Committee of the Institute of Higher Nervous Activity and Neurophysiology of RAS. Male Wistar rats (6–10 weeks old) were anesthetized by sevoflurane and decapitated. Brains were quickly submerged in ice-cold dissection solution (concentrations in mM: 124 NaCl, 3 KCl, 1.25 NaH_2_PO_4_, 26 NaHCO_3_, 0.5 CaCl_2_, 7 MgCl_2_, and 10 D-glucose, pH equilibrated with 95% O_2_–5% CO_2_). Parasagittal hippocampal slices (400 µm) were prepared using a vibratome (Leica VT1000S, Wetzlar, Germany) and immediately transferred to a recording solution (ACSF, composition as above, except the CaCl_2_ and MgCl_2_ concentrations were adjusted to 2.5 and 1.3 mM, respectively). Slices were heated to 34 °C in a water bath for 40 min and then kept at room temperature. Amyloid Aβ_25-35_ (H1192, Bachem, Bubendorf, Switzerland) was prepared 24 h before the slice incubation from 1 mM stock solution in mQ water, adjusting to 50 nM of final concentration in the ACSF. Hippocampal slices were incubated for 1 h with or without specific blocker, depending on the experimental groups.

### 4.2. Electrophysiology

During the experiments, slices were perfused by a continuous ACSF flow (appr. 4 mL/min) at 32–33 °C. Electrophysiological recordings were carried out using a SliceMaster system (Scientifica, Uckfield, UK). Field excitatory postsynaptic potentials (fEPSPs) were recorded from *Stratum radiatum* in the CA1 area using glass microelectrodes (1–2 MΩ) filled with the chamber solution. Baseline synaptic responses were evoked by paired-pulse stimulation of the Schaffer collaterals with 50 ms interval at 0.033 Hz with a bipolar electrode. Test stimulation intensity was adjusted, to evoke fEPSPs with amplitude 50% of maximal, and was kept constant throughout the experiment. Long-term potentiation (LTP) was induced with four 100 Hz trains spaced 5 min apart, as in [98]. The data were recorded and analyzed by Spike2 and SigmaPlot 11.0 (Systate Software Inc., Chicago, IL, USA). For statistical analysis, the first 3 min after tetanization (0–3 min, early LTP) and 3 min at 178–180 min after LTP induction (late LTP) were used. For baseline responses, the amplitudes and appropriate fEPSP slopes during test stimulation were evaluated. Paired-pulse facilitation (PPF) ratio was calculated as PPF = (S_2_EPSP/S_1_EPSP), where S_1_EPSP and S_2_EPSP are the slopes of fEPSP in responses to the first and the second stimuli, respectively. PPF measures were carried out just before (pre-tetanus) and after LTP recordings (post-tetanus).

### 4.3. NO Imaging of Hippocampal Slices

Parasagittal slices (400 μm), after 1 h incubation in the 95% O_2_/5% CO_2_ perfused ACSF at 34 °C, were transferred into the light-shielded camera containing continuously aerated ACSF with 5 μM of the NO-sensitive dye—DAF-FM (4-Amino-5-methylamino-2′,7′-Difluorofluorescein) diacetate. After 1 h incubation with the dye, the slices were washed in the ACSF. Further, slices were placed to experimental chamber and exposed to the drug treatment and/or tetanus. Control slices were in parallel placed in aerated ACSF without any influences. At the end of the experiment, slices were fixed in 4% formaldehyde in PBS solution for 20 min. Fixation and storage of slices occurred in light-shielded plates. Next, the slices were investigated for NO levels using a fluorescence microscope, Keyence BZ-9000 (Keyence, Osaka, Japan), equipped with mercury lamp, objective lenses (×2, ×10, ×20), excitation filter (450–490), dichroic mirror, and emission filter (520–540). Optical images were recorded and further analyzed by BZ-Analyzer (Keyence, Japan) and ImageJ (NIH, NY, USA). For semi-quantitative estimation of the NO production, a ratio of digitized dye signals was used in the region of interest (ROI) to the DAF fluorescence in the *Stratum oriens*, a hippocampal layer having weak NO synthesis activity.

### 4.4. Phosphatase Assay

Serine/threonine phosphatase activity was determined by using a nonradioactive molybdate dye-based phosphatase assay kit (Promega, Madison, WI, USA), in accordance with the instructions of the manufacturer, as reported in [99]. Following 20 min after tetanus of control or drug-treated slices (total time of the tested drug exposure to hippocampal slices in aerated ACSF is 40 min), slices were harvested and homogenized, 2 slices per 0.5 mL of buffer containing (in mM): sucrose, 250; β-mercaptoethanol, 15; EDTA, 0.1; phenylmethylsulfonyl fluoride, 0.1; TRIS-HCl, 50 (pH 7.4). Free phosphate was removed from the lysate supernatants using a Sephadex G-25 resin spin column. Phosphatase reactions were assessed in 50 μL samples at 34 °C, in buffer containing 50 mM imidazole, 0.2 mM EGTA, 0.02% β-mercaptoethanol, and 0.1 mg/mL bovine serum albumin (pH 7.2), based on the dephosphorylating rate of the synthetic 754 Da phosphopeptide RRA(pT)VA, a substrate for serine/threonine phosphatases. Specificity of the phosphatase reaction was tested by okadaic acid (100 nM) or cyclosporin A (5 μM), using PP1/PP2A and PP2B inhibitors, respectively. The reaction was stopped by addition of the dye-additive mixture (50 μL), and samples were incubated to color development for 20 min at room temperature. Absorbance was measured at 640 nm using an Infinite 200 Pro plate reader (Tecan, Salzburg, Austria). For normalization of phosphatase activity, we also measured total protein amounts in hippocampal lysates using BCA Protein Assay Kit (Thermofisher, Waltham, MA, USA), in accordance with the recommendations of the manufacturer. Protein-dependent reduction of Cu^2+^ to Cu^+^ in an alkaline medium, followed by the bicinchoninic acid sensitive detection of Cu^+^ ions, was determined in hippocampal samples, using bovine serum albumin (Amresco, Solon, OH, USA) as a calibration standard (125–2000 µg/mL). Then, 50 µL samples were incubated in the well of microplates at 37 °C for 30 min to color development and were measured by Infinite 200 Pro plate reader (Tecan, Austria) at 562 nm.

### 4.5. Isolation and Purification of Mitochondria

All manipulations with animals, before the isolation of the organs, were performed in accordance with the Helsinki Declaration of 1975 (revised in 1983), national requirements for the care and use of laboratory animals, and protocol 9/2020 of 17 February 2020, approved by the Commission on Biological Safety and Bioethics at the ITEB RAS. Male Wistar rats (2–3 months) were sacrificed after anesthesia with CO_2_. Rat brain mitochondria (RBM) were isolated as described in [100], with minor modifications. The homogenization medium contained 320 mM sucrose, 10 mM Tris (pH adjusted to 7.4 with Trizma Base), 0.5 mM EGTA, and 0.5 mM EDTA. The brain was homogenized in a Potter homogenizer (800 rpm, 20 passages). The homogenate was centrifuged twice for 4 min at 2000 *g*, and each time the precipitate was discarded. A “crude” mitochondrial pellet containing synaptosomes, myelin, and non-synaptic RBM was obtained by sedimentation from supernatant at 12,500 g for 11 min. Then, RBM was purified by the sedimentation (17,500 g for 11 min) in a discontinuous Percoll gradient (3%, 10%, 15%, and 24% Percoll in the isolation medium). The fraction of nonsynaptic mitochondria was collected and washed once in an isolation medium, free of EGTA (11,500 g, 11 min). The concentration of isolated RBM was approximately 20 mg protein/mL. Mitochondrial protein was assayed by the Biuret method, using BSA as a standard [101]. All measurements were performed at 30 °C in the standard KCl-based medium (KCl-BM: 120 mM KCl, 20 mM sucrose, 2 mM KH_2_PO_4_, and 2 mM MgCl_2_; 10 mM HEPES (pH 7.3), substrates: 5 mM pyruvate + 5 mM malate or 5 mM succinate + 1 μM rotenone, respectively). Other experimental details are given in the figures and figure legends.

### 4.6. Ca^2+^-Dependent Swelling of Mitochondria

The opening of mitochondrial permeability transition pore (mPTP) in isolated mitochondria was registered as the initiation of EGTA- and CsA-sensitive high-amplitude swelling. Mitochondrial swelling was determined by measuring a decrease in optical density at the 540 nm in mitochondrial suspension using a plate reader (Infinite 200 Tecan, Salzburg, Austria) and 96-well plates. Other details are given in the figures and figure legends.

### 4.7. ROS Detection

The rate of superoxide anion (O_2_^−•^) production was assessed using a highly sensitive chemiluminescent probe, MCLA (2-methyl-6-(p-methoxyphenyl)-3,7-dihydroxyimidazole [1,2-α]pyrazin-3-on) [102]. The kinetics of MCLA-derived chemiluminescence (MDCL) were recorded using a plate reader, Infinite 200 (Tecan, Austria). Each value on the curve is the mean ± S.E.M. of three independent measurements of luminescence, expressed in arbitrary units.

### 4.8. Measurements of the Mitochondrial Transmembrane Potential, ΔΨ_m_

∆Ψ_m_ across the inner mitochondrial membrane (IMM) was measured using an Infinite 200 plate reader (Tecan, Austria). For fluorescent measurements, incubation media contained 330 nM rhodamine 123 (excitation 480 nm, emission 525 nm). For negative control probes, we used 2-[2-[4-(trifluoromethoxy)phenyl]hydrazinylidene]-propanedinitrile (FCCP), a potent mitochondrial oxidative phosphorylation uncoupler. Other experimental details are given in the figure and table legends.

### 4.9. Drugs

Bovine serum albumin (BSA), FCCP, 4-(2-hydroxyethyl)piperazine1-ethanesulfonic acid (HEPES) and salts for solutions, rhodamine 123, rotenone, sucrose, succinate, Trizma Base, mannitol, malate, MCLA, DMSO, cyclosporin A, sephin1, cantharidin, and CGP37157 were purchased from the Sigma-Aldrich Corporation (St. Louis, MO, USA). Ethylene glycol-bis(2-aminoethylether)-N,N,N’,N’-tetraacetic acid (EGTA) was from PanReac ApppliChem (cat.№ A0878-0025, Russia). Okadaic acid was purchased from Tocris (Bristol, UK). Both 3Br-7NI and PTIO were from Enzo Life Science (New York, NY, USA). Fluorescent dye DAF-FM diacetate and DAPI were purchased from Molecular Probes (New York, NY, USA).

### 4.10. Statistical Analysis

Results are presented as mean ± standard error (S.E.M.) of *n* number of slices, from at least three different animals (*n* = 3–8, depending on the experimental series). All statistical tests were performed using SigmaPlot 11.0 (Systat Software Inc, San Jose, CA, USA) or using SPSS Statistics software (version 21, IBM Corp., Armonk, NY, USA). Significance of changes in the fEPSP characteristics, the DAF-FM fluorescence, and the phosphatase activity was tested by ANOVA with the Bonferroni test. Phosphatase reaction values are expressed as percentage (%) from control measurements (taken as 100%) and normalized to the total protein amounts in the sample. Mitochondrial measurements were made in three repeats from different animals. All tests used were two-sided; *p* < 0.05 was predetermined as defining statistically significant differences and in figures denoted by * or ^#^ for multiple comparisons.

## 5. Conclusions

The combination of the electrophysiological recordings of fEPSPs together with the direct measurements of STPP activity and NO production, as well as the monitoring of the physiological parameters in the suspension of the brain mitochondria, allows for deeper understanding of the Aβ_25-35_-mediated LTP suppression in the hippocampus. Aβ_25-35_ aggregates lead to activation of stress-induced PP1α phosphatase, shifting the kinase–phosphatase balance during the LTP induction protocol, toward the enhancement of STPP. An increase in STPP activity was well-correlated with the fEPSP suppression as well as with stimulation of the nNOS-derived NO production in the CA1, CA3, and DG layers. Furthermore, the Aβ_25-35_-dependent influence involved stimulation of the Ca^2+^ released from mitochondria through the mNCX. The results obtained in our experiments suggest the involvement of PP1α, nNOS, and mNCX in the realization of Aβ_25-35_-driven signaling in the hippocampus. These data provide at least three potential strategies for the compensation of Aβ_25-35_-mediated synaptic insufficiency, involving the blockade of the stress-induced GADD34/PP1α complex, and/or inhibition of mNCX, and targeting NO production via the nNOS blockade. Additionally, the management of the kinase/phosphatase balance through a reduction in STPP activity and/or a strengthening of serine/threonine kinase activity can be useful in preventing the Aβ_25-35_-associated impairment of hippocampal neural network functionality.

## Figures and Tables

**Figure 1 ijms-23-11848-f001:**
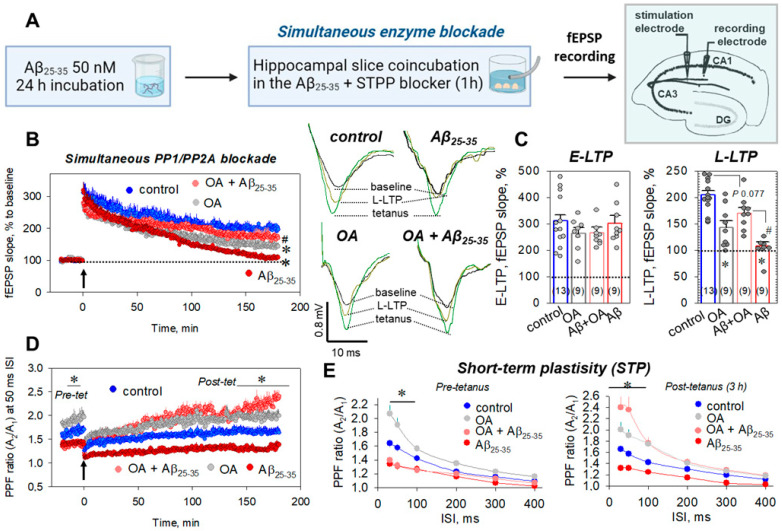
PP1/PP2A blockade under the Aβ_25-35_-dependent impairment of hippocampal synaptic plasticity. (**A**) Time protocol for preparing of Aβ_25-35_-treated slices and recording of fEPSPs responses. (**B**) Averaged curves for control slices (blue circles, *n* = 13), okadaic acid (OA)-treated slices (100 nM, gray circles, *n* = 9), Aβ_25-35_-treated slices (dark red circles, *n* = 9), and OA+ Aβ_25-35_-treated slices (red circles, *n* = 9). Arrow at 0 min indicates start of tetanic stimulation, and dotted line 100% represents normalized fEPSPs corresponding to the pre-tetanic level. Right panels display typical fEPSP responses of studied groups for pre-tetanic levels (baseline), at the start of post-tetanic recordings (tetanus) and 180 min after LTP induction (L-LTP). (**C**) Summarized statistics for (**B**) curves during early phase (E-LTP, 0–3 min) and late phase (L-LTP, 178–180 min) of long-term potentiation. * denotes *p* ˂ 0.05 for comparing group with control; # indicates *p* ˂ 0.05 for comparison of groups between themselves. The number of independent experiments is indicated in parentheses. (**D**) Paired-pulse facilitation (PPF) curves at the 50 ms ISI, corresponding to the (**B**) recordings. PPF ratios reflect short-term plasticity (STP), relating to the residual Ca^2+^ in the presynaptic endings, which increases fEPSPs in response to the second stimulus of the same strength. (**E**)—PPF ratios for different interstimulus intervals (30–400 ms) before tetanic stimulation (left panel, pre-tetanus) and for L-LTP, 3 h after tetanus (right panel, post-tetanus).

**Figure 2 ijms-23-11848-f002:**
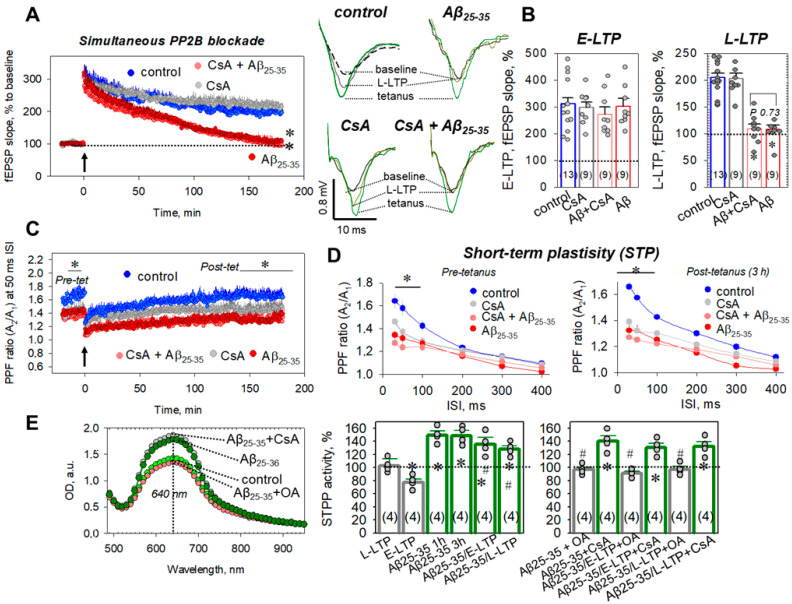
The PP2B blockade during the Aβ_25-35_-dependent impairment of hippocampal synaptic plasticity. (**A**) Averaged curves for control slices (blue circles, *n* = 13), cyclosporin A (CsA)-treated slices (5 μM, gray circles, *n* = 9), Aβ_25-35_-treated slices (dark red circles, *n* = 9), and CsA+ Aβ_25-35_-treated slices (red circles, *n* = 9). Arrow at 0 min indicates start of tetanic stimulation; dotted line 100% represents normalized fEPSPs corresponding to the pre-tetanic level. Right panels display typical fEPSP responses of studied groups for pre-tetanic levels (baseline), at the start of post-tetanic recordings (tetanus) and 180 min after LTP induction (L-LTP). (**B**) Summarized statistics for (**A**) curves during early phase (E-LTP, 0–3 min) and late phase (L-LTP, 178–180 min) of long-term potentiation. * denotes *p* ˂ 0.05 for comparing the group with control; # indicates *p* ˂ 0.05 for comparison of groups between themselves. (**C**) Paired-pulse facilitation (PPF) curves at the 50 ms ISI corresponding to the (**A**) recordings. (**D**) PPF ratios for different interstimulus intervals (30–400 ms) before tetanic stimulation (left panel, pre-tetanus) and for L-LTP, 3 h after tetanus (right panel, post-tetanus). (**E**) Optical density spectra for control (green circles) and Aβ_25-35_-treated samples (dark green circles), as well Aβ_25-35_ + OA (red circles) and Aβ_25-35_ + CsA-treated (gray circles) samples, normalized to the total protein levels. Right panels indicate summarized statistics for serine/threonine phosphatase activity (STPP) in tested groups. * denotes *p* ˂ 0.05 for comparing the group with control; # indicates *p* ˂ 0.05 for comparison of groups between themselves. The number of independent experiments corresponding to different animals is indicated in parentheses.

**Figure 3 ijms-23-11848-f003:**
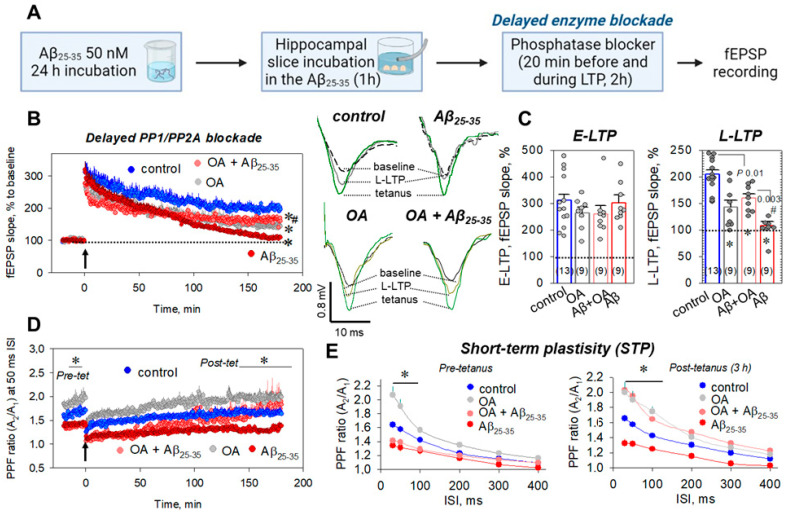
The delayed PP1/PP2A blockade during the Aβ_25-35_-dependent impairment of hippocampal synaptic plasticity. (**A**) Time protocol for preparing the Aβ_25-35_-treated slices and recording of fEPSPs responses. (**B**) Averaged curves for control slices (blue circles, *n* = 13), okadaic acid (OA)-treated slices (100 nM, gray circles, *n* = 9), Aβ_25-35_-treated slices (dark red circles, *n* = 9), and OA + Aβ_25-35_-treated slices (red circles, *n* = 9). Arrow at 0 min indicates start of tetanic stimulation; dotted line 100% represents normalized fEPSPs corresponding to the pre-tetanic level. Right panels display typical fEPSP responses of studied groups for pre-tetanic levels (baseline), at the start of post-tetanic recordings (tetanus) and 180 min after the LTP induction (L-LTP). (**C**) Summarized statistics for (**B**) curves during early phase (E-LTP, 0–3 min) and late phase (L-LTP, 178–180 min) of long-term potentiation. * denotes *p* ˂ 0.05 for comparing of the group with control; # indicates *p* ˂ 0.05 for comparison of groups between themselves. The number of independent experiments is indicated in parentheses. (**D**) Paired-pulse facilitation (PPF) curves at the 50 ms ISI corresponding to the (**B**) recordings. (**E**) PPF ratios for different interstimulus intervals (30–400 ms) before tetanic stimulation (left panel, pre-tetanus) and for L-LTP, 3 h after tetanus (right panel, post-tetanus).

**Figure 4 ijms-23-11848-f004:**
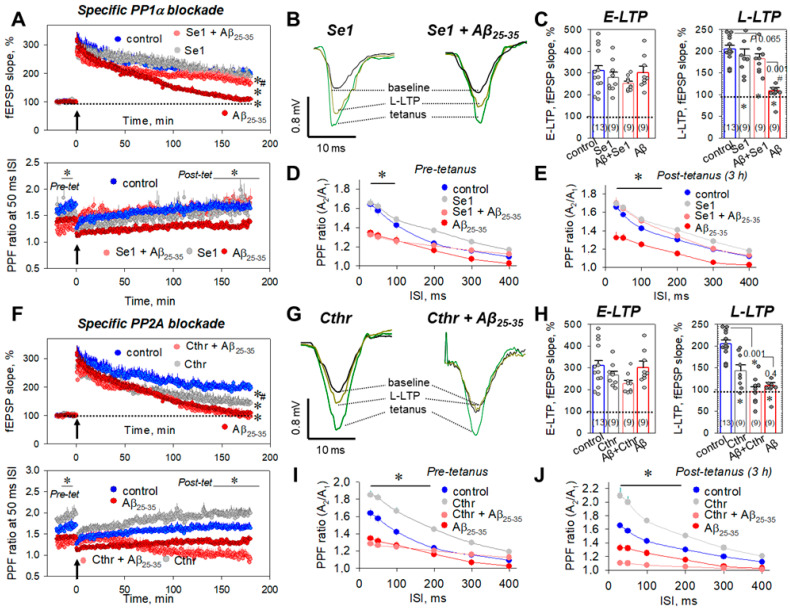
Selective PP1α or PP2A blockade during the Aβ_25-35_-dependent impairment of hippocampal synaptic plasticity. (**A**) Averaged curves for control slices (blue circles, *n* = 13), sephin1 (Se1)-treated slices (10 μM, gray circles, *n* = 9), Aβ_25-35_-treated slices (dark red circles, *n* = 9), and Se1 + Aβ_25-35_-treated slices (red circles, *n* = 9). Arrow at 0 min indicates start of tetanic stimulation; dotted line 100% represents normalized fEPSPs corresponding to the pre-tetanic level. Bottom panel represents paired-pulse facilitation (PPF) curves at the 50 ms ISI. (**B**) Typical fEPSP responses of studied groups for pre-tetanic levels (baseline), at the start of post-tetanic recordings (tetanus) and 180 min after LTP induction (L-LTP). (**C**) Summarized statistics for (**A**) curves during early phase (E-LTP, 0–3 min) and late phase (L-LTP, 178–180 min) of long-term potentiation. * denotes *p* < 0.05 for comparing the group with control; # indicates *p* < 0.05 for comparison groups between themselves. The number of independent experiments corresponding to different animals is indicated in parentheses. PPF ratios for different interstimulus intervals (30–400 ms) before tetanic stimulation ((**D**), pre-tetanus) and for L-LTP, 3 h after tetanus ((**E**), post-tetanus). (**F**) Final curves for control slices (blue circles, *n* = 13), cantharidin (Cthr)-treated slices (10 μM, gray circles, *n* = 9), Aβ_25-35_-treated slices (dark red circles, *n* = 9), and Cthr + Aβ_25-35_-treated slices (red circles, *n* = 9). Arrow at 0 min indicates start of tetanic stimulation; dotted line 100% represents normalized fEPSPs corresponding to the pre-tetanic level. Bottom panel represents paired-pulse facilitation (PPF) curves at the 50 ms ISI. (**G**) Typical fEPSP responses of studied groups for pre-tetanic levels (baseline), at the start of post-tetanic recordings (tetanus) and 180 min after LTP induction (L-LTP). (**H**) Summarized statistics for (**F**) curves during early phase (E-LTP, 0–3 min) and late phase (L-LTP, 178–180 min) of long-term potentiation. * denotes *p* < 0.05 for comparing of the group with control; # indicates *p* < 0.05 for comparison of groups between themselves. The number of independent experiments corresponding to different animals is indicated in parentheses. PPF ratios for different interstimulus intervals (30–400 ms) before tetanic stimulation ((**I**), pre-tetanus) and for L-LTP, 3 h after tetanus ((**J**), post-tetanus).

**Figure 5 ijms-23-11848-f005:**
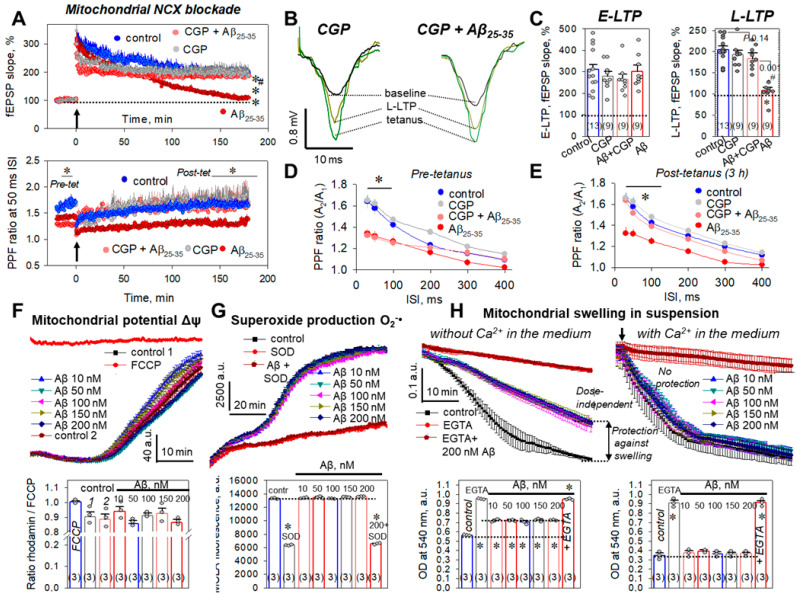
Specific mNCX blockade during the Aβ_25-35_-dependent impairment of hippocampal synaptic plasticity and Aβ_25-35_ influence on the functional parameters of brain mitochondrial suspension. (**A**) Averaged curves for control slices (blue circles, *n* = 13), CGP37157-treated slices (10 μM, CGP, gray circles, *n* = 9), Aβ_25-35_-treated slices (dark red circles, *n* = 9), and CGP37157 + Aβ_25-35_-treated slices (red circles, *n* = 9). Arrow at 0 min indicates start of tetanic stimulation; dotted line 100% represents normalized fEPSPs corresponding to the pre-tetanic level. Bottom panel represents paired-pulse facilitation (PPF) curves at the 50 ms ISI. (**B**) Typical fEPSP responses of studied groups for pre-tetanic levels (baseline), at the start of post-tetanic recordings (tetanus) and 180 min after LTP induction (L-LTP). (**C**) Summarized statistics for (**A**) curves during early phase (E-LTP, 0–3 min) and late phase (L-LTP, 178–180 min) of long-term potentiation. * denotes *p* ˂ 0.05 for comparing of the group with control; # indicates *p* < 0.05 for comparison of groups between themselves. The number of independent experiments corresponding to different animals is indicated in parentheses. PPF ratios for different interstimulus intervals (30–400 ms) before tetanic stimulation ((**D**), pre-tetanus) and for L-LTP, 3 h after tetanus ((**E**), post-tetanus). (**F**) Curves of detection of mitochondrial transmembrane potential Δ*Ψ_m_* using rhodamine 123 fluorescence in the RBM suspension (substrates: malate 5 mM + pyruvate 5 mM, top panel) and summarized statistics (bottom panel) for three independent measurements. (**G**) Curves for superoxide O^2−•^ production in mitochondrial suspension (substrates: succinate 5 mM + rotenone 1 μM) using chemiluminescence probe MCLA (top panel) and summarized statistics (bottom panel) for three independent measurements. * denotes statistically significant differences for negative control in the presence of superoxide dismutase utilizing superoxide O^2−•^ (SOD, 100U/mL) and SOD + 200 nM Aβ_25-35_. (**H**) Aβ_25-35_ influence on the swelling of brain mitochondrial suspension (substrates: succinate 5 mM + rotenone 1 μM) in the presence of 50 μM Ca^2+^ (right panels, time for addition of Ca^2+^ is reflected by an arrow) or without it (left panels). EGTA (1 mM), established Ca^2+^-chelating agent, was used as negative control, strongly preventing the swelling of mitochondria. Number of independent experiments is indicated in parentheses; * denotes *p* ˂ 0.05 in comparison to the control.

**Figure 6 ijms-23-11848-f006:**
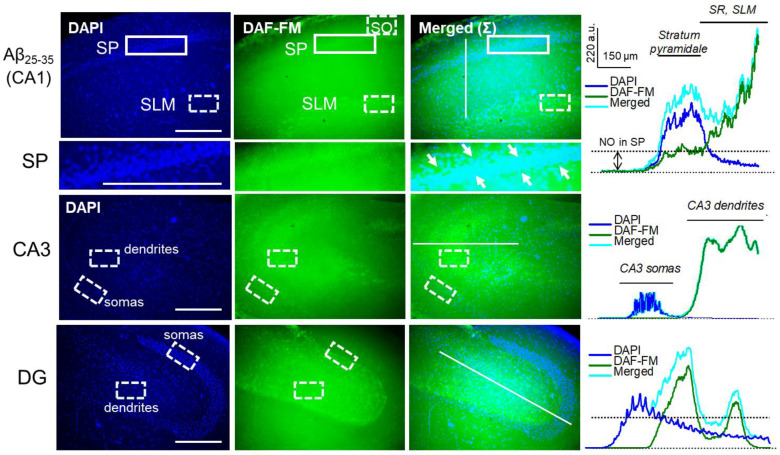
NO staining by DAF-FM dye in the Aβ_25-35_-treated hippocampal slices reveals a potent NO production in key hippocampal layers, such as CA1, stratum pyramidale (SP, arrows), stratum radiatum (SR), stratum lacunosum-moleculare (SLM), CA3, and dentate gyrus (DG). Blue fluorescence channel indicates staining of cell nuclei by DAPI. Scale bars 300 μm. Right panels indicate curves for fluorescent channel profiles, corresponding to the lines in merged (Σ) channel. SO—stratum oriens.

**Figure 7 ijms-23-11848-f007:**
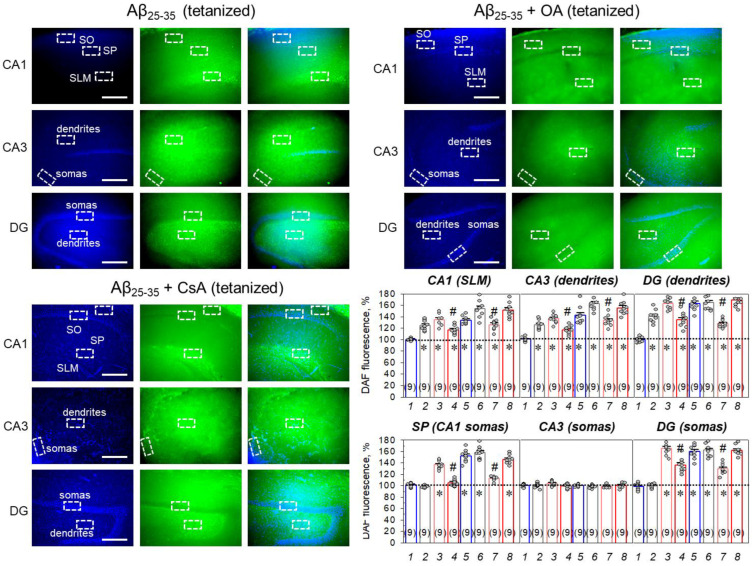
NO staining by DAF-FM dye in the tetanized Aβ_25-35_-treated slices without or in the presence of STPP blockers, okadaic acid (OA, 100 nM), or cyclosporin A (CsA, 5 μM). SP—stratum pyramidale, SLM—stratum lacunosum-moleculare (SLM), DG—dentate gyrus. Blue fluorescence channel indicates staining of cell nuclei by DAPI. Scale bars 300 μm. In histograms, summarized statistics are presented for DAF-fluorescence in dendrite-containing regions of CA1, CA3, and DG layers (top panels) as well in somas of CA1, CA3, and DG layers (bottom panels). Groups in diagrams: 1—control; 2—tetanus; 3—Aβ_25-35_; 4—Aβ_25-35_ + OA; 5—Aβ_25-35_ + CsA; 6—Aβ_25-35_ + tetanus; 7—Aβ_25-35_ + tetanus + OA; 8—Aβ_25-35_ + tetanus + CsA. Number of independent slices is indicated in parentheses; * denotes *p* ˂ 0.05 for comparison to the control; and #—*p* ˂ 0.05, when groups are compared between themselves.

**Figure 8 ijms-23-11848-f008:**
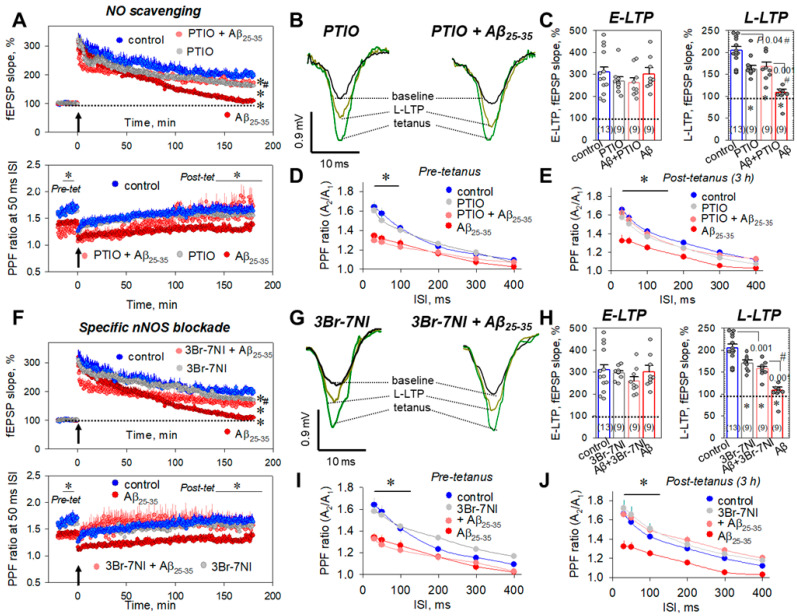
Selective disruption of NO signaling during the Aβ_25-35_-dependent impairment of hippocampal synaptic plasticity. (**A**) Averaged curves for control slices (blue circles, *n* = 13), PTIO-treated slices (50 μM, gray circles, *n* = 9), Aβ_25-35_-treated slices (dark red circles, *n* = 9), and PTIO + Aβ_25-35_-treated slices (red circles, *n* = 9). Arrow at 0 min indicates start of tetanic stimulation; dotted line 100% represents normalized fEPSPs corresponding to the pre-tetanic level. Bottom panel represents paired-pulse facilitation (PPF) curves at the 50 ms ISI. (**B**) Typical fEPSP responses of studied groups for pre-tetanic levels (baseline), at the start of post-tetanic recordings (tetanus) and 180 min after LTP induction (L-LTP). (**C**) Summarized statistics for (A) curves during early phase (E-LTP, 0–3 min) and late phase (L-LTP, 178–180 min) of long-term potentiation. * denotes *p* < 0.05 for comparing of the group with control; # indicates *p* < 0.05 for comparison of groups between themselves. The number of independent experiments is indicated in parentheses. PPF ratios for different interstimulus intervals (30–400 ms) before tetanic stimulation ((**D**), pre-tetanus) and for L-LTP, 3 h after tetanus ((**E**), post-tetanus). (**F**) Final curves for control slices (blue circles, *n* = 13), 3Br-7NI-treated slices (5 μM, gray circles, *n* = 9), Aβ_25-35_-treated slices (dark red circles, *n* = 9), and 3Br-7NI + Aβ_25-35_-treated slices (red circles, *n* = 9). Arrow at 0 min indicates start of tetanic stimulation; dotted line 100% represents normalized fEPSPs corresponding to the pre-tetanic level. Bottom panel represents paired-pulse facilitation (PPF) curves at the 50 ms ISI. (**G**) Typical fEPSP responses of studied groups for pre-tetanic levels (baseline), at the start of post-tetanic recordings (tetanus) and 180 min after LTP induction (L-LTP). (**H**) Summarized statistics for (**F**) curves during early phase (E-LTP, 0–3 min) and late phase (L-LTP, 178–180 min) of long-term potentiation. * denotes *p* < 0.05 for comparing of the group with control; # indicates *p* < 0.05 for comparison of groups between themselves. The number of independent experiments is indicated in parentheses. PPF ratios for different interstimulus intervals (30–400 ms) before tetanic stimulation ((**I**), pre-tetanus) and for L-LTP, 3 h after tetanus ((**J**), post-tetanus).

**Figure 9 ijms-23-11848-f009:**
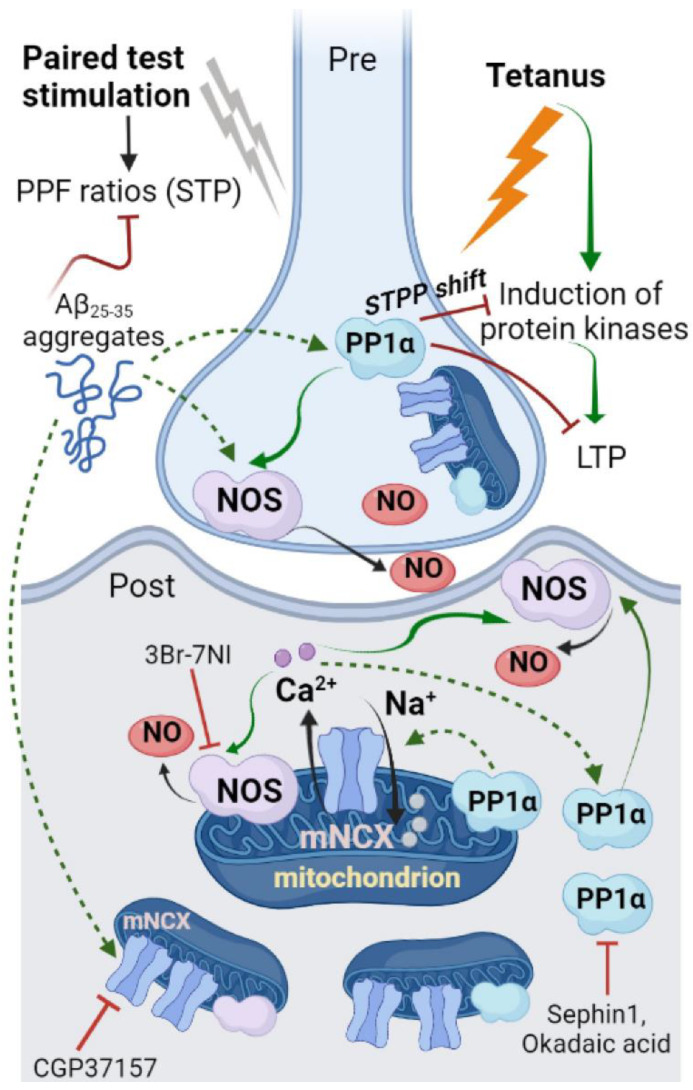
Suggested scheme for Aβ_25-35_-driven neurochemical events in the hippocampal synapses. Aβ_25-35_ aggregates (in)directly induced GADD34 and PP1α phosphatase complex activation. PP1α can dephosphorylate the nNOS, leading to disinhibition of nNOS activity followed by NO production. NO acts both pre- and postsynaptically, reducing the number of mediator vesicles and nitrosylation of protein targets, such as ion channels, enzymes, etc. Furthermore, Aβ_25-35_ aggregates stimulate the mNCX transport, elevating [Ca^2+^]_in_ in the cytosol. Increase in [Ca^2+^]_in_ can additionally facilitate the nNOS activity, through a Ca^2+^-calmodulin binding site at the nNOS. Moreover, elevation of [Ca^2+^]_in_ can provide additional GADD34 induction and enhancement of PP1α phosphatase activity. PP1α in the postsynaptic cells interferes with the kinase-mediated signals (STPP shift), preventing fEPSP maintenance after LTP induction. PP1α in the presynapse may limit vesicular exocytosis of neuromediator, impairing short-term plasticity (STP). Red blunt arrows indicate the target inhibition; sharp green arrows designate (in)direct activation of targets.

## Data Availability

The data presented in this study are available on request from the corresponding author.

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
