# Peer review of "Amyloid Aβ25-35 Aggregates Say ‘NO’ to Long-Term Potentiation in the Hippocampus through Activation of Stress-Induced Phosphatase 1 and Mitochondrial Na+/Ca2+ Exchanger"

_ijms, 2022, doi:10.3390/ijms231911848_

Round 1
Reviewer 1 Report
I read with great interest the Manuscript titled “Amyloid Aβ25-35 aggregates say ‘NO’ to long-term potentiation 2 in the hippocampus through activation of stress-induced phos-3 phatase 1 and mitochondrial Na+/Ca2+ exchanger” which falls within the aim of International Journal of Molecular Sciences. In my honest opinion, the topic is interesting enough to attract the readers' attention and methodology appropriate. I have no major concerns. I suggest authors to delete the highlights and format the bibliography according to the MDPI guidelines.
Author Response
We are grateful to the Editors and Reviewers for their thorough consideration of the manuscript and appropriate remarks, which we clarified in the revised manuscript.
Reviewer: 1
I read with great interest the Manuscript titled “Amyloid Aβ25-35 aggregates say ‘NO’ to long-term potentiation 2 in the hippocampus through activation of stress-induced phos-3 phatase 1 and mitochondrial Na+/Ca2+ exchanger” which falls within the aim of International Journal of Molecular Sciences. In my honest opinion, the topic is interesting enough to attract the readers' attention and methodology appropriate. I have no major concerns. I suggest authors to delete the highlights and format the bibliography according to the MDPI guidelines.
We deleted highlights and formatted references in the manuscript according to the IJMS guidelines.
Reviewer 2 Report
The article by Maltsev et al. studies the factors responsible for synaptic dysfunctions often observed during neurodegenerative diseases. The authors focused on Aβ peptide induced suppression of long-term potentiation (LTP) in hippocampal neurons. The authors demonstrated that aggregates of a small Aβ peptide (Aβ25-35) could induce LTP and then perform a series of analysis for understanding the mechanism. The authors tested several inhibitors for identifying the key players in the step wise suppression of LTP in neurons by the Aβ peptide. The study is done extensively, and the data reported in the manuscript provides crucial insights on the understanding of amyloid induced neuronal dysfunctions. I only have some minor suggestions that might be helpful before the manuscript is accepted for publication.
Comment 1. The authors should include the justification of utilizing the shorter version (25-35) Aβ peptide in their study. Especially when the disease associated Aβ-42 and Aβ-40 are easily available why did the authors chose to study a shorter peptide?
Comment 2. The authors need to support their notion that they tested amyloid aggregates of Aβ peptide. What did the authors do for maintaining the homogeneity (to some extent at least) of the type of aggregates being added on cells? This is crucial since there are differences in fibrils and oligomers on neuronal LTP dysfunction. The authors need to add the high-end microscopy images and fibril kinetics for the confirming the identity of aggregate type they tested. I am not sure if concentrations as low as 50 nm would be sufficient for inducing fibrillation?
Comment 3. The authors keep mentioning the effect of Aβ25-35 in their manuscript, but they actually tested the aggregates and not the intact peptide. They even conclude at few places that their observations may be amyloid induced (eg: line 187-188). This issue needs rectification.
Comment 4. The authors mention indirect influences of the Aβ25-35 aggregates to presynapse. Please Explain and add more information.
Comment 5. The authors need to add more information to support their observation that mNCX and PP1 α events occur independently and are not correlated.
Author Response
We are grateful to the Editors and Reviewers for their thorough consideration of the manuscript and appropriate remarks, which we clarified in the revised manuscript.
Reviewer 2
1) The authors should include the justification of utilizing the shorter version (25-35) Aβ peptide in their study. Especially when the disease associated Aβ-42 and Aβ-40 are easily available why did the authors chose to study a shorter peptide?
Neurotoxicity of the Aβ25-35 peptide in our experiments was obviously confirmed by an ability to dramatically suppress the synaptic transmission in the late-LTP phase after 1 h incubation in the presence of Aβ25-35 aggregates (Fig.1,2,3,4,5,8). The reinstatement of the fEPSP slopes values 3h after LTP induction to the pre-tetanic levels suggest that hippocampal synaptic plasticity in the CA3-CA1 synapses was transiently violated (Fig.1B,C, 2A,B). Indeed, it is a well established fact, that the short peptide Aβ25-35 leads to the same impairments of synaptic transmission as the full the Aβ1-42 version [Millucci et al., 2009; Balezza-Tapia et al., 2010; Pena et al., 2010]. ‘Aβ25-35 is the shortest peptide sequence that retains biological activity comparable with that of full-length Aβ1-42 and exhibits a large β-sheet aggregated structures [Pike et al. 1995; D’Ursi et al. 2004, Milucci et al., 2009]. Moreover, the Aβ25-35 peptide is present in senile plaques and degenerating hippocampal neurons in AD brains, but not in the age-matched control subjects. Certain forms of Aβ1-40 can be converted to Aβ25-35 peptide by brain proteases’ [Kubo et al. 2002, Milucci et al., 2009]. Thus, a validity of Aβ25-35 for neurobiological tasks which are related to the amyloidosis impairment modeling is not in doubt.
2) The authors need to support their notion that they tested amyloid aggregates of Aβ peptide. What did the authors do for maintaining the homogeneity (to some extent at least) of the type of aggregates being added on cells? This is crucial since there are differences in fibrils and oligomers on neuronal LTP dysfunction. The authors need to add the high-end microscopy images and fibril kinetics for the confirming the identity of aggregate type they tested. I am not sure if concentrations as low as 50 nm would be sufficient for inducing fibrillation?
Unfortunately, we did not measure the quantitative composition for Aβ25-35 aggregate solution in this work. At the same time, many studies extensively used the amyloid aggregate solutions prepared (24-48 h) from monomers [Balezza-Tapia et al., 2010; Pena et al., 2010; Mayordomo-Cava et al., 2015]. Furthermore, we tested a freshly dissolved Aβ25-35 peptide (50 nM) in respect to the hippocampal synaptic plasticity. In Suppl. Fig. 5 is shown that freshly prepared Aβ25-35 monomers actually did not impair the fEPSP responses, that is well correlated to results in papers from other authors, suggesting that amyloid monomers per se are not neurotoxic for synaptic transmission [Milucci et al., 2009; Giuffrida et al., 2009; Zhao et al., 2021]. It is well established that only amyloid oligomers have neurotoxic properties for synaptic vesicle exocytosis [Kitamura and Kubota, 2010; Zhao et al., 2021]. Furthermore, in this paper we used a more general term ‘aggregates’ because the form of these aggregates (plaques, fibrils, others) should be estimated in separate research. We suggest that the homogeneity of amyloid aggregates in our experiments was provided by a clear time protocol for preparation, the other things being equal. The existing data concerning the formation of amyloid fibrils is related to non-physiological conditions. At the same time, the composition of solution is crucial for amyloid aggregation kinetics [Milucci et al., 2009; Kitamura and Kubota, 2010]. Here, we focused on the amyloid aggregate’s influences on the parameters of the fEPSP. However, there are plans to evaluate the effects of aggregates of different sizes on synaptic transmission in a future work.
3) The authors keep mentioning the effect of Aβ25-35 in their manuscript, but they actually tested the aggregates and not the intact peptide. They even conclude at few places that their observations may be amyloid induced (eg: line 187-188). This issue needs rectification.
Absolutely correct comment. Indeed, we estimated the Aβ25-35 aggregate effects. Thus, we corrected ‘amyloid-induced LTP suppression’ to ‘amyloid aggregate-induced LTP suppression’ throughout the text.
4) The authors mention indirect influences of the Aβ25-35 aggregates to presynapse. Please Explain and add more information.
Paired-pulse facilitation (PPF) ratios reflect a short-term plasticity (STP) related to the residual Ca2+ in the presynaptic endings which increases the fEPSPs in response to the second test stimulus of the same strength at certain time intervals between first and second test stimuli. Thus, changes in the fEPSP ratios are widely considered to be related to presynaptic influences. In our experiments, Aβ25-35 aggregates decreased the PPF ratios that indicates the Aβ25-35 action was at least partly presynaptic, in addition to the suppression of postsynaptic transmission estimated by the fEPSP slope changes. We suggest that during the incubation of hippocampal slices in the presence of Aβ25-35 aggregates, there is a depletion of the pool of neurotransmitters, which leads to the decrease of PPF ratios. Citations were added.
5) The authors need to add more information to support their observation that mNCX and PP1 α events occur independently and are not correlated.
We suggest that at least the mNCX activation and PP1α induction can occur independently. In the experiments using suspension of brain mitochondria, the Aβ25-35 protect mitochondria against the swelling, effect that we relate to the mNCX activation. At the same time, the PP1α-mediated regulation of mNCX apparently is needed to integrate the cytosol-mitochondria interrelations, because most of the PP1α protein belongs to the cytosol fraction. Interestingly, between the mNCX and PP1α inhibitors in respect to the fEPSP slope changes is a clear positive correlation (Suppl. Fig. 6). Such correlation could be explained by an existence of positive feedback between the mNCX-dependent increase of the Ca2+ levels in cytosol which could stimulate induction of the PP1α activity (indirect arrows in the Fig. 9 in the manuscript). However, strictly speaking, this correlation does not allow to evaluate the sequence of target stimulation. We agree that these observations require more research. The text was improved.
References
Balleza-Tapia H, Huanosta-Gutiérrez A, Márquez-Ramos A, Arias N, Peña F. Amyloid β oligomers decrease hippocampal spontaneous network activity in an age-dependent manner. Curr Alzheimer Res. 2010 Aug;7(5):453-62. doi: 10.2174/156720510791383859.
D'Ursi AM, Armenante MR, Guerrini R, Salvadori S, Sorrentino G, Picone D. Solution structure of amyloid beta-peptide (25-35) in different media. J Med Chem. 2004 Aug 12;47(17):4231-8. doi: 10.1021/jm040773o.
Giuffrida ML, Caraci F, Pignataro B, Cataldo S, De Bona P, Bruno V, Molinaro G, Pappalardo G, Messina A, Palmigiano A, Garozzo D, Nicoletti F, Rizzarelli E, Copani A. Beta-amyloid monomers are neuroprotective. J Neurosci. 2009 Aug 26;29(34):10582-7. doi: 10.1523/JNEUROSCI.1736-09.2009.
Kitamura A, Kubota H. Amyloid oligomers: dynamics and toxicity in the cytosol and nucleus. FEBS J. 2010 Mar;277(6):1369-79. doi: 10.1111/j.1742-4658.2010.07570.x.
Kubo T, Nishimura S, Kumagae Y, Kaneko I. In vivo conversion of racemized beta-amyloid ([D-Ser 26]A beta 1-40) to truncated and toxic fragments ([D-Ser 26]A beta 25-35/40) and fragment presence in the brains of Alzheimer's patients. J Neurosci Res. 2002 Nov 1;70(3):474-83. doi: 10.1002/jnr.10391.
Mayordomo-Cava J, Yajeya J, Navarro-López JD, Jiménez-Díaz L. Amyloid-β(25-35) Modulates the Expression of GirK and KCNQ Channel Genes in the Hippocampus. PLoS One. 2015 Jul 28;10(7):e0134385. doi: 10.1371/journal.pone.0134385.
Millucci L, Raggiaschi R, Franceschini D, Terstappen G, Santucci A. Rapid aggregation and assembly in aqueous solution of A beta (25-35) peptide. J Biosci. 2009 Jun;34(2):293-303. doi: 10.1007/s12038-009-0033-3.
Peña F, Ordaz B, Balleza-Tapia H, Bernal-Pedraza R, Márquez-Ramos A, Carmona-Aparicio L, Giordano M. Beta-amyloid protein (25-35) disrupts hippocampal network activity: role of Fyn-kinase. Hippocampus. 2010 Jan;20(1):78-96. doi: 10.1002/hipo.20592.
Pike CJ, Walencewicz-Wasserman AJ, Kosmoski J, Cribbs DH, Glabe CG, Cotman CW. Structure-activity analyses of beta-amyloid peptides: contributions of the beta 25-35 region to aggregation and neurotoxicity. J Neurochem. 1995 Jan;64(1):253-65. doi: 10.1046/j.1471-4159.1995.64010253.x.
Zhao H, Huang X, Tong Z. Formaldehyde-Crosslinked Nontoxic Aβ Monomers to Form Toxic Aβ Dimers and Aggregates: Pathogenicity and Therapeutic Perspectives. ChemMedChem. 2021 Nov 19;16(22):3376-3390. doi: 10.1002/cmdc.202100428.